# An incandescent metasurface for quasimonochromatic polarized mid-wave infrared emission modulated beyond 10 MHz

Léo Wojszvzyk[1,2], Anne Nguyen[1,2], Anne-Lise Coutrot[1], Cheng Zhang [1], Benjamin Vest [1] & Jean-Jacques Greffet [1✉]

Incandescent sources such as hot membranes and globars are widely used for mid-infrared spectroscopic applications. The emission properties of these sources can be tailored by means of resonant metasurfaces: control of the spectrum, polarization, and directivity have been reported. For detection or communication applications, fast temperature modulation is desirable but is still a challenge due to thermal inertia. Reducing thermal inertia can be achieved using nanoscale structures at the expense of a low absorption and emission cross-section. Here, we introduce a metasurface that combines nanoscale heaters to ensure fast thermal response and nanophotonic resonances to provide large monochromatic and polarized emissivity. The metasurface is based on platinum and silicon nitride and can sustain high temperatures. We report a peak emissivity of 0.8 and an operation up to 20 MHz, six orders of magnitude faster than commercially available hot membranes.

[1] Université Paris-Saclay, Institut d'Optique Graduate School, CNRS, Laboratoire Charles Fabry, 91127 Palaiseau, France. [2]These authors contributed equally: Léo Wojszvzyk, Anne Nguyen. ✉email: jean-jacques.greffet@institutoptique.fr

The mid-wavelength infrared (MWIR) absorption spectrum is a material fingerprint so that MWIR spectroscopy plays a key role in many applications[1] such as chemical analysis, astrophysics, gas sensing, or security. MWIR also provides new opportunities for the development of free space communication robust to scattering and thermal imaging[2]. For all these applications, compact, robust, and inexpensive MWIR sources are needed. Light emitting diodes (LEDs) are a natural candidate but their efficiency in the MWIR is much lower than in the visible due to the $\omega^3$ dependence of the spontaneous-emission decay rate. The current state of the art[3,4] of the wall-plug efficiency in the [3–4] μm spectral range is on the order of $10^{-3}$. Quantum cascade lasers and optical parametric oscillators are bright infrared sources which can reach large modulation frequencies but are expensive. The only available compact and cheap MWIR sources are incandescent emitters such as hot membranes and globars[5]. With the advent of nanophotonics, it has become possible to design directional sources[6,7], quasi-monochromatic emitters in the near field[8–10] and in the far field[11–13], incandescent sources with high efficiency[14–16] and to combine directivity and optimized spectrum[17]. Recent reviews summarize the state of the art[18–20]. The remaining challenge is fast modulation.

Emissivity modulation has been investigated[21–28] including the demonstration of electrical modulation up to 600 kHz with stacks of quantum wells[20] operating at 200 °C. For these sources, the temperature is assumed to be uniform in the incandescent source so that Kirchhoff's law can be used to design the source.

An alternative to emissivity modulation is temperature modulation. Thermal emission by hot electrons has been observed in the visible[29–31] using metallic nanostructures and in the near-infrared (NIR)[32–34] using graphene. This process is potentially ultrafast as the electrons thermalize by interaction with phonons in the substrate on a picosecond time scale. However, due to their small emitting area, the power emitted in the MWIR by these sources is too low to be detected. By contrast, modulation of MWIR emission by hot electrons in large quantum wells[35] has been observed up to 500 kHz.

Other emitters have been reported where the electrons are in equilibrium with phonons so that the time response is governed by heat diffusion and no longer by electron-photon interaction. By using a small hot volume deposited with a good thermal contact on a cold substrate which behaves as a thermal sink, it is possible to ensure a fast thermal relaxation[36–39].

In this paper, we introduce a metasurface architecture designed to emit at a specific wavelength in the MWIR atmosphere transparency window and with a specific linear polarization using incandescence. We use materials such as platinum and silicon nitride (SiN$_x$) that can sustain heating up to at least 650 °C and can be operated during long periods of time (over 200 h) without experiencing degradation. We report thermal emission from a MWIR source with a linearly polarized emissivity reaching 0.8 at 5.1 μm with a 1.5 μm FWHM spectral width. We have measured the modulation up to our detector's cutoff frequency (20 MHz), six orders of magnitude larger than the modulation frequency of commercially available hot membranes.

## Results

**Emitter design and fabrication.** We aim at modulating the flux emitted by an incandescent emitter by modulating its temperature beyond 10 MHz. Commercially available sources such as hot membranes and globars suffer from one main limiting factor: thermal inertia restricts their modulation to a few tens of Hz[40,41].

To address this issue, we envision heating by Joule effect a metallic film with thickness $h_{metal}$ deposited on a cold substrate in a regime where the electrons and the phonons are in equilibrium.

The temperature dynamics is limited by the thermal diffusion time through the metal given by $h_{metal}^2/D$ where $D$ is the metal diffusivity on the order of $10^{-5}\,\mathrm{m^2 s^{-1}}$. A thickness of 50 nm, therefore, yields a cooling time on the order of 0.25 ns. With such a strategy in mind, the emitter temperature is no longer uniform nor stationary so that we cannot use Kirchhoff's law of thermal radiation to design the emitter. To address this issue, one can use a generalized form of Kirchhoff's law accounting for local thermodynamic equilibrium. The power radiated by the emitter, denoted by $P$, at time $t$ at a given wavelength $\lambda$ and in a direction specified by a unit vector $\mathbf{u}$ can be written as an integral over the volume of the emitter[42]:

$$P(\mathbf{u}, \lambda, t) = \sum_{\ell=\mathrm{TE,TM}} \int \eta^{(\ell)}(\mathbf{u}, \mathbf{r}, \lambda) \frac{I_{BB}[\lambda, T(\mathbf{r}, t)]}{2} d^3\mathbf{r} \quad (1)$$

where $t$ is time, $\mathbf{r}$ denotes the position of an emitting volume element of the emitter, $\eta^{(\ell)}(\mathbf{u}, \mathbf{r}, \lambda)$ is the emitter local polarized emissivity density and $I_{BB}(\lambda, T)$ is the intensity of a blackbody at temperature $T$ and wavelength $\lambda$. It has been shown[42] that the emissivity density is equal to the local absorption rate of an incident plane wave with same direction, frequency, and polarization, a relation which is a local form of Kirchhoff's law. Note that if the temperature is uniform, Eq. (1) can be cast in the form $P(\mathbf{u}, \lambda, t) = A(\mathbf{u}, \lambda) I_{BB}[\lambda, T(\mathbf{r}, t)] L_{sample}^2 \cos(\theta)$ where $A(\mathbf{u}, \lambda) = [A^{(\mathrm{TE})}(\mathbf{u}, \lambda) + A^{(\mathrm{TM})}(\mathbf{u}, \lambda)]/2$ is the total emissivity of the sample, $L_{sample}^2$ the area of the square emitter and $\theta$ the angle between emission direction and the direction normal to the sample surface. In what follows, we will deal with a quasi-Lambertian emitter and drop the angle dependence of the emissivity. Using the local form of Kirchhoff's law, it is possible to analyze incandescent emission by bodies with arbitrary shapes and arbitrary temperature distribution. Another issue to be addressed is the trade-off between emitted power which increases with the emitter size and fast thermal response which requires tiny emitters. Since the blackbody intensity is given by Bose–Einstein statistics, the brightness of a thermal source is lower than light emitted by diodes or lasers. In other words, the number of photons per mode is limited by Bose–Einstein statistics to values typically lower than 1 whereas it can be much larger than 1 for lasers. As a result, increasing the emitted power by an incandescent source requires to increase both its emissivity and its area. From that perspective, single-layer graphene emitters suffer from intrinsic low emissivity[38] (2% in the visible and the near-infrared and lower in the MWIR). A 10% absorptivity has been demonstrated using multilayer graphene[43]. Ultrafast modulation in metallic constrictions[29–31] intrinsically suffer from their nanoscale emitting area.

In order to achieve perfect absorption by a thin object, we rely on the principle of the Salisbury screen[44]: a thin metallic film placed in vacuum can absorb up to 50% of an incoming plane wave at normal incidence, if an impedance-matching condition is fulfilled. In order to achieve 100% absorption, a perfect mirror can be added below the film so that the fields reflected by the film and by the mirror interfere destructively at a wavelength $\lambda$. When illuminated by a plane wave at the same wavelength, all the incident power is then absorbed in the metallic film. Consistent with the local Kirchhoff's law, when the metallic film is heated at temperature $T$, it emits radiation upwards with an emissivity approaching 1 at wavelength $\lambda$. Remarkably, the impedance matching condition is achieved for metallic wires (MWs) thicknesses which are on the nanoscale regime.

In summary, the use of a Salisbury screen allows both optimization of the emissivity and control of the emission spectrum. However, this system would emit unpolarized light. In

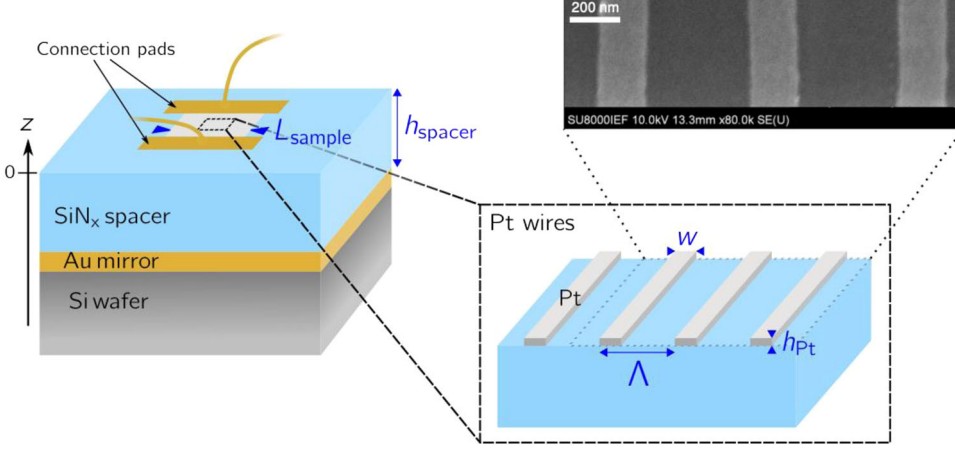

**Fig. 1 Design of the emitter.** Schematic view of the device. A periodic array of platinum wires ($w = 190$ nm, $h_{Pt} = 25$ nm, $\Lambda = 595$ nm) is deposited on a SiN$_x$ spacer ($h_{spacer} = 1.15$ μm) above a gold mirror. Electrical contacts of the platinum wires are made with two gold electrodes. The device area is $L_{sample}^2 = 100 \times 100$ μm$^2$ and is directly prepared on a silicon wafer. The array of metallic wires is designed to have an electrical impedance of 50 Ω and an optical absorptivity of 0.8 at 5.1 μm. Top right: SEM picture of some platinum wires. Scale bar, 200 nm.

order to control the emitted light polarization, we can replace the thin film with a grating consisting of a periodic array of MWs. The period is smaller than $\lambda/2$ so that there are no diffraction orders and the grating behaves as an effective planar metasurface which controls the emitted light polarization. In TE polarization, when the electric field is parallel to the MWs, such a grating on top of a perfect mirror can achieve 100% absorption and has a negligible absorptivity for TM polarization. Finally, we point out that when designing the source, we need to make sure that the electrical impedance of the emitter is 50 Ω in order to optimize the power transfer to the load and avoid signal distortions.

To implement these ideas, we use a metallic mirror deposited on a silicon substrate (see Fig. 1). A transparent dielectric spacer is then added to control the distance between the mirror and the periodic array of MWs. In our spectral range of interest (3–5 μm), silicon nitride (SiN$_x$) has a low absorption and can be heated up to at least[5] 650 °C. The geometry of the emitter was determined by optimizing the absorption in the MWs and the electrical resistance. After fabrication, the periodic array of platinum wires has a thickness of $h_{Pt} = 25$ nm and a period of $\Lambda = 590$ nm, each MW having a width $w = 190$ nm. The device total area is $L_{sample}^2 = 100 \times 100$ (μm)$^2$. The appropriate SiN$_x$ thickness is 1.15 μm. Platinum is chosen because it matches a large number of requirements: (i) it can sustain high temperatures, (ii) it is compatible with SiN$_x$ for fabrication purposes, (iii) its refractive index in the mid-infrared is compatible with a nanoscale absorber as discussed above, (iv) the electrical properties of platinum make it possible to design a device with an electrical impedance approaching 50 Ω, matching the electrical source impedance, and (v) the conductivity of platinum depends on temperature and this feature can be conveniently used as a real-time thermometer. The price to pay is that the impedance matching condition is not perfectly fulfilled for all temperatures. Details on the numerical optimization and fabrication steps are given in the Methods section.

**Dynamics characterization.** To get a first insight of the emitter dynamic response, we carried out a time-resolved emission measurement of the response under a short gaussian voltage pulse with FWHM of 10 μs. Figure 2a shows the emitted power detected by a fast MWIR detector (MCT, Kolmar) and the simulated signal using the measured temperature of the metallic

wires (see Methods). The emission intensity temporal profile follows the voltage waveform. We measure a 10–90% rise and fall time slightly shorter than 7 μs, which is limited by the voltage pulse duration. The observed delay in the rising phase, also observed in the temperature measurement, is attributed to the thermal inertia of the SiN$_x$ layer lying under the MWs. Indeed, the pulse duration is larger than the diffusion time over the spacer layer which is on the order of $h_{SiN_x}^2/D_{SiN_x} \sim 10^{-6}$ s for 1D-heat conduction. Nonetheless, this time-resolved measurement shows that our emitter responds in less than 7 μs which would mean that it can sustain modulation frequencies up to at least $10^5$ Hz.

In order to investigate more precisely the emitter response to frequency modulation, we build the setup sketched in Fig. 2b and detailed in the Methods section. A sinusoidal voltage with an amplitude of 20 V$_{pp}$ oscillating at frequency $f_{elec} = \omega_{elec}/(2\pi)$ is applied to the device. The expression of the local temperature can be cast in the form:

$$T(\mathbf{r}, t) = T_0 + \Delta T_{DC}(\mathbf{r}) + \Delta T_{AC}(\mathbf{r}, 2\omega_{elec}) \cos(2\omega_{elec}t), \quad (2)$$

where $T_0$ is the reference temperature (taken here as equal to the room temperature). A DC offset is expected as a consequence of the RMS power that is delivered to the device as well as an AC component varying at frequency $f_{th} = 2f_{elec}$. We record the polarized emitted signal as a function of frequency for both TE and TM polarizations. The experimental results are presented in Fig. 2c.

As expected, TE emission (Fig. 2c, blue plot) due to the MWs is the leading contribution and three different frequency regimes are observed. They are delimited with colored backgrounds in Fig. 2c: a low-frequency regime up to ~100 kHz (blue hue) where the signal depends very weakly on $f_{th}$, a higher frequency regime from ~100 kHz to ~10 MHz (pink hue) with a $1/\sqrt{\omega_{th}}$ decrease and a cutoff at 20 MHz (no color). We checked that this cutoff can be attributed to the MCT detector that acts as a low-pass filter.

## Discussion

**Frequency regimes and efficiency.** In order to analyze the different frequency regimes of the emitted signal, we use the generalized Kirchhoff's relation. The TE-polarized emitted power at

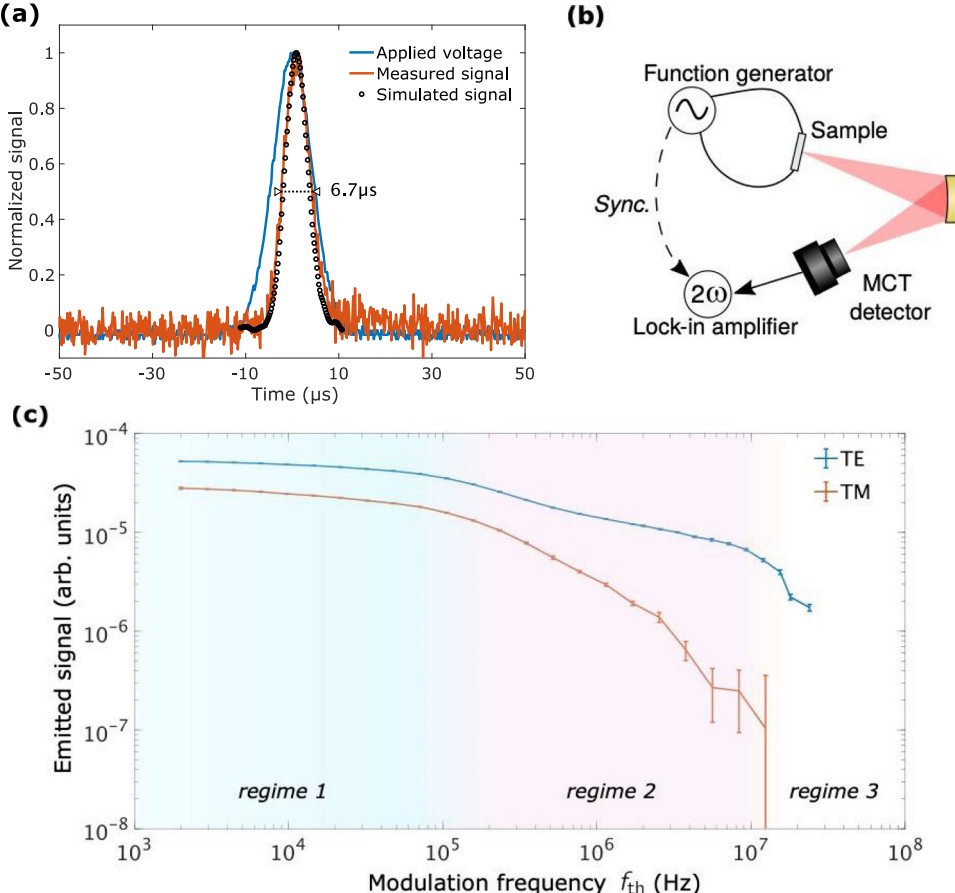

**Fig. 2 Dynamic characteristics of the emitter. a** Emission pulse when applying a gaussian voltage. The FWHM of the voltage (blue) and emitted (orange) pulses are respectively 10 μs and 6.7 μs. The voltage is normalized by its maximum value 9.7 V, the measured optical signal is normalized by its maximum value (corresponding to 4.4 μW total emitted power). The optical signal was simulated (black dots) by computing $I_{BB}[\lambda_0, T_0 + \Delta T(t)] - I_{BB}[\lambda_0, T_0]$ where $\lambda_0 = 5.1$ μm, $T_0 = 295$ K is the ambient temperature and $\Delta T(t)$ is the measured temperature increase of the metallic wires (not shown, see Methods for calibration details). The maximum temperature $T_0 + \Delta T(t)$ used for normalization of the simulated optical signal is 570 K, for which a maximum power contrast of 4.4 μW is achieved. **b** Schematic view of the experimental setup to characterize the frequency response of the emitter. A function generator is used to apply a sine voltage across the sample, a fast MCT detector collects the emitted intensity and a lock-in amplifier extracts the $2\omega_{elec}$ component of the detected signal. **c** Measured frequency response of the emitter. The scale shows the thermal frequency $f_{th} = 2f_{elec}$. The error bars indicate standard deviation.

wavelength $\lambda$ can be cast in the form:

$$P^{(TE)}(\lambda, t) = \int \frac{\eta^{(TE)}(\mathbf{r}, \lambda)}{2} I_{BB}[\lambda, T_0 + \Delta T_{DC}(\mathbf{r}) + \Delta T_{AC}(\mathbf{r}, 2\omega_{elec}) \cos(2\omega_{elec}t)] d^3\mathbf{r}$$

(3)

If $\Delta T_{AC} \ll T_0 + \Delta T_{DC}$, a Taylor expansion of $I_{BB}$ with regards to temperature can be made. As our focus is drawn on the power radiated by the fundamental mode of the modulated emitter power $P^{(TE)}(\lambda, \omega_{th})$, only the odd-powers in the Taylor development of Eq. (3) will contribute, i.e. odd time-harmonics. The first order term contains the information on the spectral dependence of $P^{(TE)}(\lambda, \omega_{th})$, provided the third order contribution remains negligible:

$$P^{(TE)}(\lambda, t) = \text{cst} + \left[ \int \frac{\eta^{(TE)}(\mathbf{r}, \lambda)}{2} \frac{\partial I_{BB}}{\partial T}[\lambda, T_0 + \Delta T_{DC}(\mathbf{r})] \Delta T_{AC}(\mathbf{r}, 2\omega_{elec}) d^3\mathbf{r} \right] \times \cos(2\omega_{elec}t)$$

(4)

where $\text{cst} = \int \frac{\eta^{(TE)}(\mathbf{r}, \lambda)}{2} I_{BB}[\lambda, T_0 + \Delta T_{DC}(\mathbf{r})] d^3\mathbf{r}$ is the DC-component of the polarized emitted power. We have checked that this condition is verified in our case: the measurement of the platinum temperature gives $\Delta T_{DC} \simeq 251°C$ and $\Delta T_{AC} \simeq 132°C$ at $f_{elec} = 10$ kHz under $V_{PP} = 20$ V input voltage.

The signal emitted by the device is thus controlled by the temperature dynamics in the region where absorption takes place according to the localized Kirchhoff's law. Absorption essentially occurs in the MWs in TE polarization and in the SiN$_x$ spacer in TM polarization, resulting in a different frequency dependence as discussed in the Methods section.

To estimate the contrast in emitted power, we assume the TE-polarized emissivity to be 0.6 in a [4.5, 6.5] μm range and 0 otherwise. Using the measured temperature, we find that the resulting contrast in emitted power yields $8.9 \times 10^{-8}$ W/sr at 20 MHz and $2.9 \times 10^{-6}$ W/sr at 20 kHz. The resistance was measured to be 36 Ohm so that the electrical input power is 1.3 W. The wall-plug efficiency is thus estimated to be $2.2 \times 10^{-7}$ at 20 MHz and $6.9 \times 10^{-6}$ at 20 kHz. To the best of our knowledge, this is the fastest electrically driven MWIR source modulated through temperature variation.

**Emission spectrum.** We now turn to the analysis of the spectrum emitted by an incandescent source with modulated temperature. It is seen from Eq. (1) that the emission spectrum changes as the temperature changes with time. Hence a spectrum may only be defined if a timestamp is specified, unequivocally fixing the temperature. Therefore, a stationary emitted spectrum cannot be

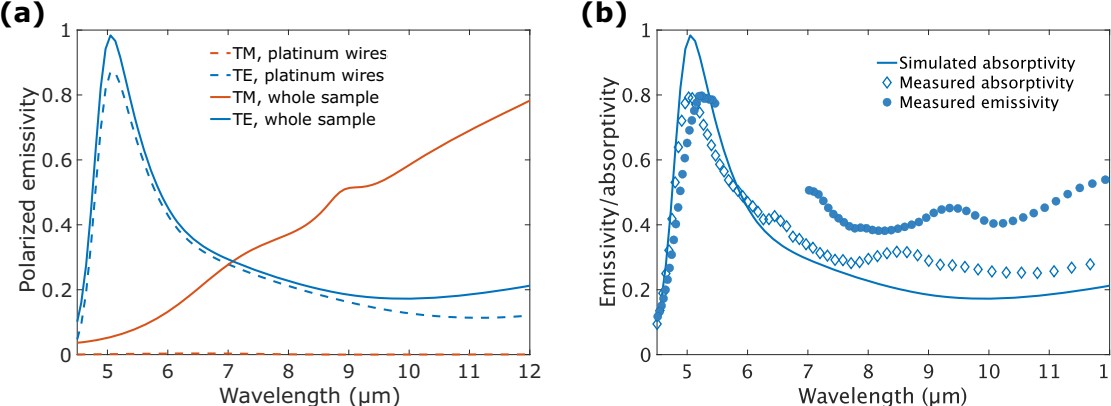

**Fig. 3 Simulated and measured emissivity spectra of the device. a** RCWA-calculated polarized emissivity spectrum of the metallic wires (dashed) and of the whole sample (thick lines) at angle 17.5°. **b** Measured emissivity spectrum. Thick line: theoretical TE polarized absorptivity of the device under 17.5° incidence simulated using RCWA. Diamonds: measured TE polarized absorptivity via reflectivity-measurement on a FTIR microscope with a 15× Cassegrain objective (NA=0.58) using a globar source and a polarizer. Filled dots: measured TE polarized emissivity spectrum rescaled such that the peak value matches the one of the absorptivity. Systematic errors due to air absorption occurred in the calibration of the spectral response of the detector so that data are removed in the [5.5 μm; 7 μm] range.

defined. However, when the first-order expansion giving Eq. (4) is valid, the spectral dependence of the fundamental mode of the modulated emitted power does not follow $A(\lambda)I_{BB}[\lambda, T(t)]$ but $A(\lambda)\partial I_{BB}[\lambda, T_0 + \Delta T_{DC}(\mathbf{r})]/\partial T$. Note that the DC temperature in the emitter may not be uniform so that spectra emitted by volume elements with different temperatures are distinct.

As explained in the introduction, the device has been designed in order to maximize the absorption at a given wavelength and for TE polarization. From Kirchhoff's law, the emissivity of the MWs is equal to the absorptivity, which can be computed numerically using Rigorous Coupled-Wave Analysis (RCWA). The numerical optimization predicts an absorptivity in TE polarization for the MWs layer of 0.88 at 5.1 μm with a spectral FWHM of 1.3 μm, as shown in Fig. 3a. As expected, the metallic sub-wavelength wires behave as a nearly perfect emitter in TE polarization whereas they are not contributing to TM emission.

Using Kirchhoff's law, we can access the total emissivity of the device by determining its absorptivity $A(\lambda)$ through a reflectivity measurement as there is no transmission through the device. The first experiment consists of using a FTIR microscope to take reflectivity data of our device illuminated by a globar source while feeding the MWs with a DC voltage so as to reproduce the temperature increase obtained in the modulated regime. A reflectivity spectrum from the gold contacts was taken as a reference. Results are presented in Fig. 3b with diamond markers. From Kirchhoff's law, the absorptivity spectrum gives direct access to the total spectral emissivity of our device.

The second experiment consists of measuring the device emissivity spectrum directly. For this purpose, we modulate the input voltage at $f_{elec} = 10$ kHz. Emitted photons are collected and sent to the FTIR spectrometer using gold mirrors to avoid chromatic aberrations. The spectrometer is operated in step-scan mode. For each position of the interferometer moving mirror, we use a lock-in amplifier to extract the component at frequency $2\omega_{elec}$ from the detected signal. Results of this emissivity spectrum measurement (see Methods) are presented with filled dot markers in Fig. 3b. We focus on TE emissivity as it is predominant in the spectral region of interest. The TE emissivity displays a peak at 5.1 μm with a 1.5 μm FWHM which is essentially due to the MWs as seen in Fig. 3a. Good agreement is obtained between simulation and measurements.

In summary, we have presented the design of a compact and robust MWIR source that has been modulated up to 20 MHz and features optical properties such as spectral selectivity and linear polarization of the emitted light. The design of the source relies on a recent extension of the Kirchhoff's law to model MWIR radiation by bodies with temperatures varying spatially and temporally. Optimizing the incandescent emission requires to control concomitantly the temperature field and the electromagnetic absorption in a nanoscale volume at a given wavelength and polarization. This results in a device emitting polarized radiation with a narrow band spectral emissivity larger than 0.8 that can be easily reproduced on a large-scale using standard fabrication procedures. When applying 20 V$_{pp}$ to the device, the DC temperature increase is 251 °C and the temperature modulation at 20 kHz has a 132 °C amplitude corresponding to a contrast of the emitted modulated flux of $2.3 \times 10^{-6}$ W/sr in a band centered around 5.1 μm with a 1.5 μm FWHM spectral width. The power decays as $1/\sqrt{\omega_{th}}$ so that the emitted power is one order of magnitude smaller at 2 MHz. In terms of wall-plug efficiency, our source is comparable to available commercial LEDs (IoffeLED, LED53TO8TEC), reaching $7 \times 10^{-6}$ at a 20 kHz modulation frequency without the need for an external cooling. The efficiency can be further improved by increasing the DC and AC operating temperatures to take advantage of the non-linear dependence of the emitted signal. This can be done by increasing the applied voltage and by further optimizing the thermal management of the system. All these features indicate that incandescent metasurfaces are good candidates for MWIR emission.

## Methods

**Sample fabrication**. A 200 nm-thick gold mirror was deposited on a 2″ standard silicon wafer by classical electron-beam evaporation, with a 3 nm thick titanium adhesion layer. The 1.15 μm thick silicon nitride spacer was obtained by plasma enhanced chemical vapor deposition at 175 °C. The platinum wires were elaborated using e-beam lithography (Nanobeam NB4 lithography system). We spin-coated 200 nm of PMMA A4 resist after a 15 min hotplate prebaking at 180 °C. The device was exposed under a voltage of 80 kV and a beam current around 1.6 nA. We developed the pattern in a mixed solution of MIBK:IPA=1:3 for 40 s at room temperature, subsequently rinsing it with IPA and blowing nitrogen. A 25 nm platinum film was deposited by electron-beam evaporation and the finalized platinum wires were obtained after lift-off in acetone for 1 h. Finally, we used laser lithography (KLOE DILASE 650, 4% power, 2 mm/s writing speed) and electron-beam evaporation for lift-off process to realize two platinum/gold 0.5 mm × 0.5 mm electrodes to contact the platinum wires.

**Temperature measurements**. Temperature measurement of the platinum wires was performed taking advantage of their electrical resistance temperature-

dependence. In other words, the hot filament plays the role of a temperature sensor. By adding a known resistor in series, we are able to record voltages across both the device and the resistor (which gives the intensity) and determine the resistance of the sample as a function of time. The temperature coefficient of the MWs electrical resistance has been calibrated. The device was inserted in a cryostat ensuring isothermal experimental conditions. The electrical resistance was measured as the cryostat chamber temperature was slowly heating up to room temperature. Assuming temperature-dependence of the resistance to be $R(T) = R(T_0)[1 + \alpha(T - T_0)]$, the calibration leads to $\alpha = 1.98\,10^{-3}\,\Omega/K$ for $T_0 = 295$ K. The pulse measurement (Fig. 2a) was performed using a function generator (Aim-TTi TGP 3152) and a MCT detector (Mercury Cadmium Telluride Kolmar KMPV11-0.5-J1/DC). Both signals were observed on an oscilloscope (LeCroy WaveRunner LT26M) triggered on the voltage signal. Emission pulses were averaged over several thousands of sweeps.

**Emission dynamics.** In order to measure the frequency response of the emitter (Fig. 2c), we used a 50 MHz function generator (Aim-TTi TGP 3152) and the same MCT detector (Kolmar KMPV 11-0.5-J1) whose cutoff frequency is about 20 MHz. The sample DC resistance was measured to be 36 $\Omega$ under $20V_{PP}$ feeding so that the RMS delivered power when the oscillating amplitude voltage $U_0$=10 V is 1.3 W.

In order to isolate the amplitude of the $2\omega_{elec}$-component of the signal, we use a lock-in amplifier (Zürich Instruments HF2LI), whose cutoff frequency is 50 MHz. Above about 2 MHz, the radio waves emitted by the MWs produce an additional parasitic signal. To circumvent this issue, we shielded the MCT detector as well as the device with grounded aluminum foil acting as Faraday cages.

**Numerical modeling.** Electromagnetic computations were performed using a rigorous coupled wave analysis (RCWA) method[45]. Refractive indices of gold and platinum were found in the literature[46] and the dielectric constant of silicon nitride at ambient temperature was modeled using a 2-distribution of Drude-Lorentz oscillators, following the Brendel Borman model for amorphous materials: $\epsilon(\nu) = \epsilon_\infty + \sum_{j=1}^{2} X_j(\nu)$ where $X_j(\nu) = \frac{1}{\sqrt{2\pi}\sigma_j}\int_{-\infty}^{\infty} \exp\left[-\frac{(x-\nu_{0j})^2}{2\sigma_j^2}\right]\frac{\nu_{pj}^2}{x^2-\nu^2-i\nu\nu_{\tau_j}}dx$, $(\epsilon_\infty)$ being the high-frequency permittivity, $\nu_{0j}$ the center frequency of the distribution, $\nu_{pj}$ the plasma frequency, $\nu_{\tau j}$ the damping constant, $\sigma_j$ the standard deviation of the Gaussian distribution of oscillators. The parameters used were $\nu_{0,1} = 858.5$ cm$^{-1}$, $\nu_{\tau,1} = 89.9$ cm$^{-1}$, $\nu_{p,1} = 1284.5$ cm$^{-1}$, $\sigma_1 = 94.7$ cm$^{-1}$, $\nu_{0,2} = 2773.9$ cm$^{-1}$, $\nu_{\tau,2} = 136.4$ cm$^{-1}$, $\nu_{p,2} = 33.7$ cm$^{-1}$, $\sigma_2 = 2132.6$ cm$^{-1}$, and $\epsilon_\infty = 3.7182$. The temperature of the emitter when applying a voltage was calculated with finite element method (FEM) using a homemade library (Sim-Photonics). In order to take into account the temperature dependence of the platinum resistance, it is necessary to do a time-domain computation using a finite difference explicit scheme. Because it is very time-and-memory consuming to simulate the transient dynamics until stationary regime is reached in the case of high frequencies, we made an approximation. For frequencies higher than 100 kHz, the temperature oscillations can be considered to be small enough so that the resistance is well approximated by its average value. We then solve the problem in the harmonic regime to get directly the temperature oscillation amplitude.

**Temperature dynamics.** After linearization, the temperature dynamics rule the emitted flux dynamics. We first deal with the temperature inhomogeneity which depends a priori on $x,y$ and $z$. Using finite element methods to solve the diffusion equation in our system for modulation frequencies up to 20 MHz, we noticed that the in-plane relative change in $T_{DC}(\mathbf{r})$ over depths on the order of the thickness of our device (a few microns) was negligible. We shall then only consider the depth-dependence $T_{DC}(0,0,z)$. Furthermore, as long as modulation frequencies are small enough compared to the inverse of the heat diffusion time through the platinum height ($\sim 10^{10}$ Hz), which is the range presented in this paper, the temperature in the MWs can be considered as homogeneous. Equation (4) can then be treated with $z = 0$ so that $T_{DC,AC}^{Pt}(0, 0, z)$ can be estimated using $T_{DC,AC}^{Pt}(0, 0, 0)$. Inserting this approximation in Eq. (4) allows removing the temperature from the integral. From there, the emissivity of the MWs $A_{MW}(\lambda)$ explicitly appears as the integral of the TE-polarized emissivity density over the MWs: $A_{MW}(\lambda)L_{sample}^2\cos(\theta) = \int_{MW} \eta^{(TE)}(\mathbf{r}, \lambda)d^3\mathbf{r}$, taking advantage of the fact that the MWs are expected to essentially absorb TE-polarized light. RCWA simulations show that the TE-polarized emissivity is indeed mainly due to the platinum and that the TM-polarized one is only due to the spacer, yielding $A^{(TE)}(\lambda) \simeq A_{MW}(\lambda)$ for $\lambda \in [4.5 - 10]$ μm and $A^{(TM)}(\lambda) = A_{spacer}(\lambda)$ for $\lambda \in [4.5 - 12]$ μm as shown in Fig. 3a. Using the linearized form of the intensity introduced in Eq. (4), the $2\omega_{elec}$-component of the TE-polarized signal detected by the MCT detector, $S^{(TE)}$, can be cast in the form

$$S^{(TE)}(2\omega_{elec}) \propto \left[\int_0^\infty R(\lambda)A_{MW}(\lambda)\frac{1}{2}\frac{\partial I_{BB}}{\partial T}(\lambda, T_0 + \Delta T_{DC})d\lambda\right]\Delta T_{AC}(0, 0, 0, 2\omega_{elec}),$$

(5)

where $R(\lambda)$ is the detector response function. What stands out of this equation

is that the signal depends on the frequency only via the MWs temperature oscillation. Hence, the analysis of the frequency dependence of the signal amounts to discuss the frequency dependence of the emitter temperature. Let us compute $\Delta T_{AC}(2\omega_{elec})$. From a thermal point of view, the problem is the following: the electric current flowing through the MWs dissipates power by Joule effect. The radiated power is negligible compared to the power flowing to the substrate by conduction. Then, we need to solve the diffusion equation with a source term given by a sinusoidal flux equal to the electrical power dissipated in the emitter. In order to understand the origin of the different frequency regimes, we introduce the characteristic heat diffusion length in a medium labeled $j$ given by $\sqrt{\pi D_j/\omega_{elec}}$. When the diffusion length is much larger than the size of the emitter, the latter can be approximated by a point-like source from a thermal point of view. This condition corresponds to modulation frequencies smaller than $f_c = \frac{D_{Si}}{L_{sample}^2} \sim 10^4$ Hz (region in blue in Fig. 2c). We can then use the analytical solution of temperature in a semi-infinite medium with a point-like source at the interface[47]. We find that the temperature oscillation does not depend on frequency for a point-like source. This is in agreement with the slow power-law decay observed which corresponds to a transition regime where the device has a finite-size.

At frequencies much higher than $f_c$ (region in pink in Fig. 2c), the diffusion length becomes much smaller than the lateral size of the film. Thus, heat transfer becomes a multilayer 1D problem, which can be easily solved analytically. Solving this equation for a semi-infinite medium $j$ with thermal conductivity $K_j$ and thermal diffusivity $D_j$ heated with a uniform harmonic thermal flux $\phi_0$ imposed at its surface $z=0$, the amplitude of temperature oscillation reads[47]:

$$\Delta T_{AC}(z, \omega_{th}) = \frac{\phi_0}{\sqrt{2}K_j k_{th,j}}e^{k_{th,j}z}, z<0$$

(6)

where $k_{th,j} = \sqrt{\omega_{th}/(2D_j)}$ is the thermal propagation constant in medium $j$. This expression evaluated in the MWs at $z = 0$, combined to equation (5), shows an interesting feature: the temperature oscillation amplitude decreases as $1/\sqrt{\omega_{th}}$. As a consequence, the amplitude of the emitted signal decays with frequency. Thus, there is a trade-off between modulation frequency and signal amplitude with this type of incandescent source. Note that the $1/\sqrt{\omega_{th}}$ drop in the frequency response is predicted by this model provided that the periodic array of MWs can be modeled as a uniform plane. This assumption is valid up to a modulation frequency $\nu_{th} \sim D/\Lambda^2 \sim 10^7$ Hz where $\Lambda$ is the period of the structure and $D$ the thermal diffusivity of SiN$_x$ which is beyond our detector's cutoff.

It is interesting to analyze the behavior of the signal emitted in TM polarization (Fig. 2c, orange plot). As briefly mentioned above, in TM polarization, the emitting zone is not the MWs but the silicon nitride layer. According to Eq. (6), the amplitude of temperature oscillation depends on the depth $z$ in the SiN$_x$. As a result, the frequency dependence in TM polarization is expected to differ from TE polarization. In the 1D regime (blue region in Fig. 2c), we introduce $a_{spacer}(\lambda, z)$ by integrating the local emissivity density $\eta_{spacer}(\mathbf{r}, \lambda)$ of the SiN$_x$ over $x$ and $y$. The $2\omega_{elec}$-component of the TM-polarized signal detected by the MCT, $S^{(TM)}$, can be written most generally as

$$S^{(TM)}(2\omega_{elec}) \propto \int_{-h_{spacer}}^{0}\int_0^\infty R(\lambda)a_{spacer}(\lambda, z)\frac{1}{2}\frac{\partial I_{BB}}{\partial T}[\lambda, T_0 + \Delta T_{DC}(z)]d\lambda \times \Delta T_{AC}(z, 2\omega_{elec})dz$$

(7)

where $h_{spacer}$ is the thickness of the SiN$_x$ layer. To first approximation, we have assumed that $a_{spacer}(\lambda,z)$ does not depend on $z$ and have neglected inhomogeneity in the DC-component of temperature variation in the spacing layer. Analytical computations show that the TM-polarized signal decreases as $1/\omega_{elec}$. This model agrees with the experimental results in TM thermal emission.

**Spectrum measurement.** Spectral absorptivity measurements deduced from reflectivity (Fig. 3b) were carried out on a ThermoScientific Nicolet Fourier-Transform InfraRed (FTIR) 5700 with a 15x Cassegrain objective with NA = 0.58, operated in rapid-scan mode using a globar source and a KBr beam-splitter. The emitter was heated via ohmic losses using a DC voltage whose value was chosen as the RMS of the AC voltage used in the emission measurement.

The spectrum measurement of the emitter (Fig. 3b) was done with a Bruker Vertex V70 FTIR spectrometer, with a KBr beam-splitter and an internal MCT detector. In spite of the low emitted power, we benefited from the intrinsic modulated signal and used step-scan mode: the emitter is supplied with a voltage of 20 $V_{PP}$ at frequency $f_{elec}$, thus the temperature and the emitted radiation oscillate at frequency $2f_{elec}$. At each position of the mirror, we record only the $2\omega_{elec}$-component of the signal delivered by the detector.

In order to extract the emissivity spectrum from the emitted power measurement, we model the signal as follows. We first note that we can assume the temperature to be uniform in the top part of the device (MWs, spacer, and mirror) which emits light because we used a low modulation frequency $f_{th}$=20 kHz so that the heat dissipated in the MWs diffuses over a typical depth $\sqrt{D/f_{th}} \simeq 10$ μm. Given the amplitude of temperature elevation, the expression of the ($\ell$)-polarized

radiated spectral intensity can be linearized as

$$\frac{1}{L_{sample}^2 \cos\theta} \int \frac{\eta^{(\ell)}}{2}(\mathbf{r},\lambda) d^3\mathbf{r} \frac{I_{BB}[\lambda,T(t)]}{2} = \frac{A^{(\ell)}(\lambda)}{2} I_{BB}[\lambda,T(t)]$$

$$\simeq \frac{A^{(\ell)}(\lambda)}{2}\left(I_{BB}(\lambda,T_0+\Delta T_{DC}) + \frac{\partial I_{BB}}{\partial T}(\lambda,T_0+\Delta T_{DC})\Delta T_{AC}(2\omega_{elec})\cos(2\omega_{elec}t)\right).$$

(8)

We find that the modulated ($\ell$)-polarized signal $S^{(\ell)}(\lambda,2\omega_{elec})$ depends on $\omega_{elec}$ only through $\Delta T_{AC}(2\omega_{elec})$. Thus, the emissivity spectrum $A^{(\ell)}(\lambda)$ of the emitter can be deduced from $S^{(\ell)}(\lambda,2\omega_{elec})$ using

$$A^{(\ell)}(\lambda) \propto S^{(\ell)}(\lambda,2\omega_{elec})\left[R(\lambda)\frac{1}{2}\frac{\partial I_{BB}}{\partial T}(\lambda,T_0+\Delta T_{DC})\Delta T_{AC}(2\omega_{elec})\right]^{-1},$$

(9)

where $R(\lambda)$ is the response of the FTIR internal MCT detector. One missing step to extract the total emissivity is the knowledge of $\Delta T_{DC}$. The DC temperature elevation is obtained from the platinum temperature-dependent electrical resistance.

## Data availability
Datasets supporting the findings of this study are available in Zenodo with the identifiers https://doi.org/10.5281/zenodo.4298762 (https://zenodo.org/record/4298762).

## Code availability
The code used in this work is available in the same Zenodo repository as the datasets.

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

## Acknowledgements
We gratefully thank J.-P. Hugonin for help with the electromagnetic simulations, M. Besbes for help with thermal simulations, R. Colombelli and A. Bousseksou for providing access to their FTIR microscope reflectivity measurement setup and F. Marquier for many discussions. L.W. and A.N. acknowledge support from Direction Générale de l'Armement. This work was supported by the grant ANR-17-CE24-016 of the Agence Nationale de la Recherche. J.J.G. and C.Z. acknowledge the support of the SAFRAN-IOGS chair on Ultimate Photonics.

## Author contributions
J.-J.G. conceived and supervised the project. L.W. designed and fabricated the device. C.Z. and A.L.C. participated in the device fabrication. L.W., A.N., and B.V. designed, performed the experiments, and analyzed the data. L.W., A.N., and J.-J.G. wrote the paper with input from all authors.

## Competing interests
The authors declare no competing interests.
