## [Peer Review File · Nature Communications]

REVIEWER COMMENTS

Reviewer #1 (Remarks to the Author):

The manuscript demonstrates the fast modulation of thermal radiation by modulating the local temperature of resonant metal gratings. The demonstrated work is generally interesting but lacks clarity. Additional experiments could strengthen the demonstration.

- 1) The demonstration claims to hold the record of being the fastest modulation of temperature/thermal emission. Please check this recent paper (doi: 10.1038/s41377-019-0158-6).
- 2) The pulsed thermal radiation can arise from any small change in local temperature. However, a larger change in temperature results in a stronger pulse and is practically useful. This information is missing in experiments. Fig. 2a plots thermal emission in arbitrary units.
- 3) Local temperature change is claimed to be 70 deg C. However, there are no experimental results to support the calculations.

Reviewer #2 (Remarks to the Author):

The authors present a directly driven thermal source capable of achieving high-speed modulation up to 20MHz. The proposed structure cleverly relies on narrow and thin metallic gratings as thermal absorber/emitter. This enables the high-speed operation, and simultaneously linearly polarized emission. The metallic absorber/emitter are placed on a spacer with back reflector to increase the absorption/emissivity for a designed spectral window (5.1 μ m) achieving an emissivity of 0.8.

I believe that this work does present an interesting concept for the community with strong results. The manuscript is well written and provides thorough discussion and analysis in the main text and methods. Before publication I do have some comments/questions:

- The initial part of the introduction is missing some general references. E.g. a recent review on Mid-IR applications would be helpful to interested readers
- The authors compare their results with state-of-the art and commercial emitters. Maybe a comparative table (in method) summarizing the main performances/features would help understand the reader the strength of the presented emitter.
- What motivated the choice for SiN, besides its high thermal operation capability? Did the author consider other criteria?
- Would adding a cladding (i.e. thermal sink) help?
- The authors present the emitted signal as a function of the modulated thermal frequency. For TE polarization a signal up to 20MHz is observed. The cutoff is attributed to the MCCT detector bandwidth limit (acting as a low pass-filter). How is the MCT detector frequency response taken into account in the measurement. Is the data shown raw (not calibrated)? If that is the case, how can the author be sure that the amplitude dependences observed are not influenced by the MCT detector's response?
- If the cut-off is from the MCT detector, what is the device limit? Does the response follow $1/\sqrt{(\omega_{th})}$ as shown in method, or is there a sharp cut-off to be expected?
- High modulation speed comes with small heater. Yet, smaller heater gives smaller power. In my understanding the speed is defined by the thickness of the metallic film. Their large area can boost the power. Did the author investigate the emitted power as a function of Area? Is there a fundamental limit in area/power achievable while maintaining speed?
- What are the trade-offs considered for the thickness (with respect to impedance matching and diffusion length)?
- Is the power for TE and TM of same order of magnitude as suggested in plot Fig2c (at low speed)?
- The author measured the wall-plug efficiency. How was this estimated? Also, where does the power of 1.3W come from? Is 20Vpp and 50Ohm assumed, or what is the impedance of the device?
- What are the possible paths to make the linewidth sharper? Can the metasurface design be optimized to further enhance the spectral linewidth?

Reviewer #3 (Remarks to the Author):

The authors report on the design, fabrication and characterisation of an incandescent emitter incorporating a metasurface and Salisbury screen. They experimentally demonstrate very fast response times, and high values of emissivity at the wavelength which satisfy the impedance matching condition. The experimental work is supported by theory and modelling. Whilst there have been previous studies of similar architectures, the novelty of the work lies in the very high modulation rates achieved in the mid-infrared. I believe that the work presented is technically correct, although in places the text could be a little clearer. Some of the statements relating to previous/related work are not quite correct and should be amended (see below). I believe that the manuscript does make a significant contribution to the field. However, I feel that the authors could better justify the novelty of their work – in some senses, such high modulation rates are as-expected in such small structures. Overall, I believe that the manuscript could be suitable for publication in Nature Communications, but would ask that the authors address the following points:

(1) In first paragraph of the introduction, when discussing MWIR LEDs, the authors state that they “cannot be operated without external cooling”. I don’t think that this is strictly correct. There are many LEDs described in the literature that do not require cooling, for example the Al_xIn_{1-x}Sb based LEDs described by Nash et al [IEEE Sensors Journal 9, 1240 (2009)] had a wall plug efficiency of approximately 0.02% at room temperature, and similar LEDs have been incorporated into commercially available carbon dioxide sensors. In addition, wall plug efficiencies of approximately 0.15% were reported in interband cascade LEDs [Abell et al, Appl. Phys. Lett. 104, 261103 (2014)], also at room temperature.

(2) In the Results Section, top of page 5, the authors state that graphene emitters suffer from the “intrinsic low emissivity (2%)”. For devices based on single layer graphene this is true, but it’s been shown that multi-layer graphene can be used to create emitters with emissivity of 10% or more. (Barnard et al, Appl. Phys. Lett. 108, 131110 (2016)). These devices also had an emitting area of approximately $2.5 \times 10^5 \mu\text{m}^2$, much larger than the area of $10^3 \mu\text{m}^2$ mentioned by the authors. Please can the authors amend the text appropriately, and also add a reference to this paper.

(3) Results section, page 5, second paragraph. Please can the authors make clear what “thicknesses” are being referred to in the last sentence.

(4) When discussing the dynamic response of the emitter, as shown in Figure 2(a), there is a delay in the start of the emission relative to the voltage being applied. The authors attribute this to the thermal initial of the SiNx. However, the emission and the applied voltage both switch-off at the same time. Please can the authors describe the origins of this behaviour? From the data presented in Figure 2(a), it appears as though the duration of the emitted light pulse is shorter than the applied voltage pulse, which is somewhat counter-intuitive.

(5) Please can the authors comment on the oscillatory behaviour observed in the measured emissivity and absorptivity above wavelengths of approximately 6 μm (Figure 3b). Why do the emissivity and absorptivity have peaks at slightly different wavelengths?

(6) The authors estimate the wall-plug efficiency to be 10^{-6} at 20 kHz modulation frequency, falling to 10^{-8} at 20 MHz. They state that the efficiency can be increased by “increasing the DC and AC operating temperatures” which can be done by “increasing the applied voltage and further optimizing the thermal management of the system”. I think these approaches could potentially improve the wall-plug efficiency, but please can the authors comment on what factors might limit the wall-plug efficiency. As the impedance matching condition for the Salisbury screen is met for one wavelength, does this mean that some of the electrical energy supplied will always be lost through diffusion of heat?

Reviewer #1 (Remarks to the Author):

The manuscript demonstrates the fast modulation of thermal radiation by modulating the local temperature of resonant metal gratings. The demonstrated work is generally interesting but lacks clarity. Additional experiments could strengthen the demonstration.

Answer: We thank the reviewer for her/his positive appreciation. We had performed temperature measurements but this was unclear. We have expanded the discussion of these measurements as explained below (see the detailed answer to point 2, 3 and point 4 of referee 3). We have also performed additional temperature measurements.

1) The demonstration claims to hold the record of being the fastest modulation of temperature/thermal emission. Please check this recent paper (doi: 10.1038/s41377-019-0158-6).

Answer: The reference cited by the reviewer deals with a modulation of the emissivity obtained using an external femtosecond laser. We have added this reference in the text in the section dealing with modulated emissivity (now reference 28).

We do not claim the record of fast modulation. Note that we have cited other devices much faster than ours. Our claim is limited to the category of cheap, compact, reliable and **electrically-driven** MIR emitters whose intensity is modulated through temperature modulation and with the potential of being quasimonochromatic and linearly polarized.

Modification: 28. Xiao, Y., Charipar, N. A., Salman, J., Piqué, A. & Kats, M. A. Nanosecond mid-infrared pulse generation via modulated thermal emissivity. Light Sci. Appl. 8, 51 (2019).

2) The pulsed thermal radiation can arise from any small change in local temperature. However, a larger change in temperature results in a stronger pulse and is practically useful. This information is missing in experiments. Fig. 2a plots thermal emission in arbitrary units.

Answer: The emitted flux depends on the collection solid angle, on the emitting area and on the sample temperature and emissivity. While the emitted power spectrum was plotted in arbitrary units for the pulse illustration, the total emitted power was given when we evaluate the wall-plug efficiency from our measurements of the emissivity and the temperature. The emitted power is 9 μW at a modulation of 20 kHz when a signal with peak amplitude of 10V is applied.

There was a lack of detailed information on the temperature measurement in the previous text. We have now expanded the discussion of the metallic wire temperature measurement by taking advantage of the temperature dependence of the platinum wire itself (the heating filament is itself the temperature sensor).

Furthermore, we had been using so far the literature value of the temperature dependence of platinum resistivity. We have now calibrated the dependence of the resistance of our sample as a function of the temperature and this introduces a significant modification. This allows to obtain an instantaneous value of the temperature

by measuring the resistance in real time. We have added the following paragraph in the Methods section.

*Modification: " **Temperature measurements.** Temperature measurement of the platinum wires was performed taking advantage of its electrical resistance temperature-dependence. In other words, the hot filament plays the role of a temperature sensor. By adding a known resistor in series, we are able to record both voltages across the device and the resistor (which gives the intensity) and determine the resistance of the sample as a function of time. The temperature coefficient of the platinum wire electrical resistance has been calibrated. The device was inserted in a cryostat insuring isothermal experimental conditions. The electrical resistance was measured as the cryostat chamber temperature was slowly heating up to room temperature. Assuming temperature-dependence of the resistance to be $R(T) = R(T_0)[1 + \alpha(T - T_0)]$, the calibration leads to $\alpha = 1.98 \cdot 10^{-3} \Omega/K$ for $T_0 = 297 \text{ K}$."*

We have then repeated the pulse experiment while recording the temperature as a function of time. We present the results in Fig. 2 below (response to referee 3). The maximum temperature is 570 K. This temperature measurement has been used to simulate the shape of the emitted signal using the blackbody radiance and a curve has been added to Fig. 2 a. Knowing the emissivity, the area, the solid angle and the temperature, we find that the corresponding maximum emitted power is 4.4 μW .

3) Local temperature change is claimed to be 70 deg C. However, there are no experimental results to support the calculations.

Answer: This value was derived from a measurement of the electrical resistance and the use of the literature temperature-dependence of the resistivity. As indicated above, we have now calibrated the temperature dependence of the sample resistance and we report the result of the calibration curve in the method section. We have also reported the values of the temperature increase at 10 kHz.

Modification: "the measurement of the platinum temperature gives $\Delta T_{DC} \simeq 251^\circ\text{C}$ and $\Delta T_{AC} \simeq 132^\circ\text{C}$ at $f_{elec} = 10 \text{ kHz}$ under $V_{pp}=20 \text{ V}$ input voltage. "

Reviewer #2 (Remarks to the Author):

The authors present a directly driven thermal source capable of achieving high-speed modulation up to 20MHz. The proposed structure cleverly relies on narrow and thin metallic gratings as thermal absorber/emitter. This enables the high-speed operation, and simultaneously linearly polarized emission. The metallic absorber/emitter are placed on a spacer with back reflector to increase the absorption/emissivity for a designed spectral window (5.1 μm) achieving an emissivity of 0.8.

I believe that this work does present an interesting concept for the community with strong results. The manuscript is well written and provides thorough discussion and analysis in the main text and methods.

Answer: We thank the reviewer for her/his positive assesment.

Before publication I do have some comments/questions:

- The initial part of the introduction is missing some general references. E.g. a recent review on Mid-IR applications would be helpful to interested readers.

Answer: We have added a recent review paper on Mid-IR applications and a review on free-space communications.

Modification: "The Mid Wavelength InfraRed (MWIR) absorption spectrum is a material fingerprint so that MWIR spectroscopy plays a key role in many applications¹ such as chemical analysis, astrophysics, gas sensing or security. MWIR also provides new opportunities for the development of free space communication robust to scattering and thermal imaging². For all these applications, compact, robust and inexpensive MWIR sources are needed."

...

*1.Popa, D. & Udrea, F. Towards Integrated Mid-Infrared Gas Sensors. Sensors **19**, 2076 (2019).
2.Majumdar, A. K. & Ricklin, J. C. Free-space laser communications: principles and advances. (Springer, 2008).*

- The authors compare their results with state-of-the art and commercial emitters. Maybe a comparative table (in method) summarizing the main performances/features would help the reader understand the strength of the presented emitter.

Answer: We thank the reviewer for her/his suggestion. We have clarified the state of the art and respective performances based on the comments of Referee n°3 (points 1 & 2). The strength of our emitter is highlighted in the text as follows:

Modification: "In this paper, we introduce a metasurface architecture designed to emit at a specific wavelength in the MWIR atmosphere transparency window and with a specific linear polarization using incandescence. [...] We have measured the modulation up to our detector's cutoff frequency (20 MHz), six orders of magnitude larger than the modulation frequency of commercially available hot membranes."

"To the best of our knowledge, this is the fastest electrically driven MWIR source modulated through temperature variation."

"In terms of wall-plug efficiency, our source is comparable to available commercial LEDs (IoffeLED, LED53TO8TEC), reaching 10^{-6} at a 20 kHz modulation frequency without need for an external cooling."

Coming back to the Referee's suggestion, we do agree that a table would be helpful. We had actually done such a table when reviewing the state of the art. However, in order to be fair to all authors whose devices are under comparison, we would need to list all relevant characteristics specific to each device (e.g. type of modulation, physics at play, targeted spectral range, emitting area, emissivity or emissivity contrast, input power, useful radiated power, frequency limit, constraints with regards to operating

conditions...). Adding a compact table to our manuscript would imply selecting arbitrarily some criteria making the comparison either incomplete or unfair.

We plan to make an exhaustive comparative table in a forthcoming paper.

- What motivated the choice for SiN, besides its high thermal operation capability? Did the author consider other criteria?

Answer: There are three key issues i) transparency in the mid IR, ii) material sustaining high temperature, iii) compatibility with platinum over a large range of temperatures.

- Would adding a cladding (i.e. thermal sink) help?

Answer: The substrate is a thermal sink. A forthcoming paper will provide full details on the thermal management of the device.

- The authors present the emitted signal as a function of the modulated thermal frequency. For TE polarization a signal up to 20MHz is observed. The cutoff is attributed to the MCCT detector bandwidth limit (acting as a low pass-filter). How is the MCT detector frequency response taken into account in the measurement. Is the data shown raw (not calibrated)? If that is the case, how can the author be sure that the amplitude dependences observed are not influenced by the MCT detector's response?

Answer: The data shown are raw data. The frequency response of the detector as a function of frequency has been measured (see Figure 1). It is flat up to the cutoff frequency.

Figure 1. Detector frequency response characterization.

- If the cut-off is from the MCT detector, what is the device limit? Does the response follow $1/\sqrt{\omega_{th}}$ as shown in method, or is there a sharp cut-off to be expected?

Answer: The limit observed is due to the detector cutoff. The expected behavior is indeed given by $1/\sqrt{\omega_{th}}$. This model is valid inasmuch as the periodic array of wires can be modelled as a uniform plane which is valid up to $D/v_{th} \sim \Lambda^2$ where Λ is the period of the structure and D the thermal diffusivity of SiN_x. This corresponds to a frequency of 38 MHz using a period of 595 nm. At higher frequencies (above the bandwidth of our MCT), the response behavior may have a different frequency dependence that we have not yet investigated.

- High modulation speed comes with small heater. Yet, smaller heater gives smaller power. In my understanding the speed is defined by the thickness of the metallic film. Their large area can boost the power. Did the author investigate the emitted power as a function of Area? Is there a fundamental limit in area/power achievable while maintaining speed?

Answer: This is an important point. The emitted power is proportional to the area, the emissivity and the radiance. The cooling takes place by thermal diffusion normally to the surface. Fast cooling only requires a thin absorber and is not limited by the area. The key issue that we had to solve was to attain an absorption close to 1 with a nanometer scale absorber.

Regarding the question of the area, it is found that in the linear regime (temperature increase proportional to input power), a larger area with the same electrical power does not change the emitted power because the temperature increase is inversely proportional to the area. In the non linear regime, namely for larger temperature increase, it is favorable to have large temperatures and therefore, *smaller areas* for a given input power. A full study of the thermal management and thermal optimization an in-depth analytical treatment of the temperature field and emitted power which is beyond the scope of this paper. It will be reported elsewhere.

- What are the trade-offs considered for the thickness (with respect to impedance matching and diffusion length)?

Answer: There is no trade-off as we have enough parameters to optimize both. We need to adjust the optical impedance and the electrical impedance. Our parameters are the thickness, the filling fraction, the pitch between the MWs and the area.

- Is the power for TE and TM of same order of magnitude as suggested in plot Fig2c (at low speed)?

Answer: Yes, the polarized integrated signal only differ by a factor of 2. However, the difference in the spectral range of interest (4,5- 6,5) μm is on the order of 10. Most of the difference comes from the range 6-10 μm as seen in Fig. 3 of the manuscript.

- The author measured the wall-plug efficiency. How was this estimated? Also, where does the power of 1.3W come from? Is 20Vpp and 50Ohm assumed, or what is the impedance of the device?

Answer: The emitted power is derived from the measurement of the emissivity and the measurement of the sample temperature. We have modified the sentence as follows:

Modification: To estimate the contrast in emitted power, we assume the TE-polarized emissivity to be 0.6 in a [4.5, 6.5] μm range and 0 otherwise. Using the measured temperature, we find that the resulting contrast in emitted power yields $8.9 \cdot 10^{-8}$ W/sr at 20 MHz and $2.9 \cdot 10^{-6}$ W/sr at 20 kHz. The resistance was measured to be 36 Ohm so that the electrical input power is 1.3 W with a 10 V amplitude. The wall-plug efficiency is thus estimated to be $2.2 \cdot 10^{-7}$ at 20 MHz and $6.9 \cdot 10^{-6}$ at 20 kHz.

- What are the possible paths to make the linewidth sharper? Can the metasurface design be optimized to further enhance the spectral linewidth?

Answer: In principle, it is possible to modify the quality factor of the absorber by replacing the Salisbury absorber by a sharper resonator.

Reviewer #3 (Remarks to the Author):

The authors report on the design, fabrication and characterisation of an incandescent emitter incorporating a metasurface and Salisbury screen. They experimentally demonstrate very fast response times, and high values of emissivity at the wavelength which satisfy the impedance matching condition. The experimental work is supported by theory and modelling. Whilst there have been previous studies of similar architectures, the novelty of the work lies in the very high modulation rates achieved in the mid-infrared. I believe that the work presented is technically correct, although in places the text could be a little clearer. Some of the statements relating to previous/related work are not quite correct and should be amended (see below). I believe that the manuscript does make a significant contribution to the field. However, I feel that the authors could better justify the novelty of their work – in some senses, such high modulation rates are as-expected in such small structures.

Answer: We thank the reviewer for her/his positive assesment. We agree that the time response is expected to be short for a tiny object. It is also expected for a tiny object that it will not sustain high temperatures and that it will not emit light because of its tiny absorption cross section. We have summarized the achievements in the abstract:

Modification: "Here, we introduce a metasurface that combines nanoscale heaters to ensure fast thermal response and nanophotonic resonances to provide large monochromatic and polarized emissivity. The metasurface is based on platinum and silicon nitride and can sustain high temperatures. We report a peak emissivity of 0.8 and an operation up to 20 MHz, six orders of magnitude faster than commercially available hot membranes."

Overall, I believe that the manuscript could be suitable for publication in Nature Communications, but would ask that the authors address the following points:

(1) In first paragraph of the introduction, when discussing MWIR LEDs, the authors state that they "cannot be operated without external cooling". I don't think that this is strictly correct. There are many LEDs described in the literature that do not require cooling, for example the AlIn_{1-x}Sb based LEDs described by Nash et al [IEEE Sensors Journal 9, 1240 (2009)] had a wall plug efficiency of approximately 0.02% at room temperature, and similar LEDs have been incorporated into commercially available carbon dioxide sensors. In addition, wall plug efficiencies of approximately 0.15% were reported in interband cascade LEDs [Abell et al, Appl. Phys. Lett. 104, 261103 (2014)], also at room temperature.

Answer: We thank the referee for pointing out these references. We have modified the text accordingly.

Modification: "Light emitting diodes (LEDs) are a natural candidate but their efficiency in the MWIR is much lower than in the visible due to the ω^3 dependence of the spontaneous-emission decay rate. The current state of the art^{3,4} of the wall-plug efficiency in the [3-4] μm spectral range is on the order of 10^{-3} ."

...

*3. Abell, J. et al. Mid-infrared interband cascade light emitting devices with milliwatt output powers at room temperature. Appl. Phys. Lett. **104**, 261103 (2014).*

*4. Nash, G. R. et al. Mid-Infrared $\text{Al}_x\text{In}_{1-x}\text{Sb}$ Light-Emitting Diodes and Photodiodes for Hydrocarbon Sensing. IEEE Sens. J. **9**, 1240–1243 (2009).*

(2) In the Results Section, top of page 5, the authors state that graphene emitters suffer from the "intrinsic low emissivity (2%)". For devices based on single layer graphene this is true, but it's been shown that multi-layer graphene can be used to create emitters with emissivity of 10% or more. (Barnard et al, Appl. Phys. Lett. 108, 131110 (2016)). These devices also had an emitting area of approximately $2.5 \times 10^5 \mu\text{m}^2$, much larger than the area of $10^3 \mu\text{m}^2$ mentioned by the authors. Please can the authors amend the text appropriately, and also add a reference to this paper.

Answer: We have modified the text by emphasizing that 2% is the absorption of graphene in the visible and NIR for a monolayer but that 10% absorption has been achieved for multilayer graphene.

Modification: "From that perspective, single layer graphene emitters suffer from intrinsic low emissivity³⁸ (2% in the visible and the near infrared and lower in the MWIR). A 10% absorptivity has been demonstrated using multilayer graphene⁴³. Ultrafast modulation in metallic constrictions²⁹⁻³¹ intrinsically suffer from their nanoscale emitting area."

29. Downes, A., Dumas, Ph. & Welland, M. E. Measurement of high electron temperatures in single atom metal point contacts by light emission. *Appl. Phys. Lett.* **81**, 1252–1254 (2002).

30. Buret, M. et al. Spontaneous Hot-Electron Light Emission from Electron-Fed Optical Antennas. *Nano Lett.* **15**, 5811–5818 (2015).

31. Malinowski, T., Klein, H. R., Iazykov, M. & Dumas, P. Infrared light emission from nano hot electron gas created in atomic point contacts. *EPL Europhys. Lett.* **114**, 57002 (2016).

38. Lawton, L. M., Mahlmeister, N. H., Luxmoore, I. J. & Nash, G. R. Prospective for graphene based thermal mid-infrared light emitting devices. *AIP Adv.* **4**, 087139 (2014).

43. Barnard, H. R. et al. Boron nitride encapsulated graphene infrared emitters. *Appl. Phys. Lett.* **108**, 131110 (2016).

(3) Results section, page 5, second paragraph. Please can the authors make clear what “thicknesses” are being referred to in the last sentence.

Answer: We have clarified the text and indicated that we discuss the metallic wires thickness. (We can absorb 100 with a grating with a thickness of 25 nm and a filling fraction of 0.32.)

Modification: "Remarkably, the impedance matching condition is achieved for the metallic wires (MWs) thicknesses which are on the nanoscale regime."

(4) When discussing the dynamic response of the emitter, as shown in Figure 2(a), there is a delay in the start of the emission relative to the voltage being applied. The authors attribute this to the thermal initial of the SiNx. However, the emission and the applied voltage both switch-off at the same time. Please can the authors describe the origins of this behaviour? From the data presented in Figure 2(a), it appears as though the duration of the emitted light pulse is shorter than the applied voltage pulse, which is somewhat counter-intuitive.

Answer: It is a very interesting question. By measuring the platinum wire electrical resistance time evolution, we have deduced the temperature dynamics shown in Figure 2.

Figure 2. Time-resolved electrical measurement of the temperature elevation in the metallic wires (orange) under a gaussian pulse voltage input with $10 \mu\text{s}$ FWHM (blue).

Inserting the temperature elevation ΔT (orange plot) into Planck's law for the central wavelength of our device $\lambda_0 = 5.1 \mu\text{m}$, we recover the shape of the optical signal by plotting $I_{BB}(\lambda_0, T_0 + \Delta T) - I_{BB}(\lambda_0, T_0)$ (Figure 3). The results obtained from this independent electrical measurement is consistent with the optical measurement in Fig.2.a of the manuscript.

Figure 3. Emission pulse when applying a gaussian voltage pulse. Blue and orange: plot taken from Fig.2.a of the manuscript. Yellow: Optical signal at $5.1 \mu\text{m}$ deduced from the electrical measurement of the temperature elevation.

Figure 2 clearly shows that the temperature dynamics reproduces the voltage pulse with some retardation. The remaining question is why the temperature dynamics is different on the rising and falling edges.

Our interpretation is that at short times, the temperature depends on the cooling of the metal through thermal diffusion in the layer underneath the platinum wires made of SiN_x . At times longer than the thermal diffusion time through the SiN_x layer, the thermal behavior depends on the thermal diffusivity in the substrate made of silicon. It turns out that we use a pulse whose duration is longer than the diffusion time through the SiN_x layer so that the cooling is driven by thermal diffusion in Si. *To make a long story short, for the rising part of the pulse, the wire appears to be on SiN_x while for the cooling part of the pulse, the wire appears to be on Si.*

(5) Please can the authors comment on the oscillatory behaviour observed in the measured emissivity and absorptivity above wavelengths of approximately $6\mu\text{m}$ (Figure 3b). Why do the emissivity and absorptivity have peaks at slightly different wavelengths?

Answer: This is a question that kept us busy for some time. There are actually two different origins of the oscillations.

i) When looking at the measured absorptivity, we observe a bump at $8.5\mu\text{m}$. This bump is not observed in the simulation. We attribute this bump to the presence of NH in the material deposited. This is not accounted for in the tabulated refractive index of a pure material.

ii) When looking at the emission spectrum, we observe differences with the absorptivity. They are due to a measurement artefact when using the FTIR for the emission measurement. We operated in the step scan mode and a spectrum required several hours of recording. It turns out that the signal to noise ratio becomes too low for path differences larger than $133\mu\text{m}$. This is equivalent to introducing a window of width $2 \times 133\mu\text{m}$ on the data. We believe that this is the origin of the spurious oscillations when taking the Fourier transform. We have been able to simulate these oscillations numerically by introducing numerically such a window.

(6) The authors estimate the wall-plug efficiency to be 10^{-6} at 20 kHz modulation frequency, falling to 10^{-8} at 20 MHz. They state that the efficiency can be increased by “increasing the DC and AC operating temperatures” which can be done by “increasing the applied voltage and further optimizing the thermal management of the system”. I think these approaches could potentially improve the wall-plug efficiency, but please can the authors comment on what factors might limit the wall-plug efficiency. As the impedance matching condition for the Salisbury screen is met for one wavelength, does this mean that some of the electrical energy supplied will always be lost through diffusion of heat?

Answer: The reviewer is right. The vast majority of the electrical energy is lost in thermal conduction. This is a direct consequence of the fact that the radiated flux is orders of

magnitude smaller than the conduction flux in the substrate. Optimizing the emissivity is not enough to compete with conduction losses. Although this can be solved in DC regime by thermal insulation (suppression of conduction losses for a membrane in vacuum suspended with long thin wires), it cannot be solved in our case because we need thermal conduction to ensure fast cooling.

The trade-off between efficiency and fast modulation will be extensively discussed in a forthcoming paper.

Note that we have corrected the values of the efficiency using the corrected temperature measurements.

REVIEWERS' COMMENTS

Reviewer #1 (Remarks to the Author):

Thanks for adding the calibration details to Fig. 2 and its discussion in the manuscript.

Reviewer #2 (Remarks to the Author):

The authors have answered all my questions. I have only one final suggestion. In the answer regarding the $1/\sqrt{\omega_{th}}$ behavior, the author points out that this is valid for $D/v_{th} \sim \Lambda^2$, which in this case is beyond the detector cut-off. Still, this is an important condition (useful for device design) and should be added to the method section (line ~360).

Reviewer #3 (Remarks to the Author):

I thank the authors for comprehensively addressing the points raised by all the referees. My recommendation is that the paper is now published in Nature Communications.

REVIEWERS' COMMENTS

Reviewer #1 (Remarks to the Author):

Thanks for adding the calibration details to Fig. 2 and its discussion in the manuscript.

We are grateful for this helpful remark on the response of our voltage to a gaussian voltage pulse. Fig. 2 has been updated and contains the simulated emitted signal deduced from the temperature calibration. Details on the device resistance temperature-dependence calibration have been added in the Methods section as follows:

“Temperature measurements. Temperature measurement of the platinum wires was performed taking advantage of their electrical resistance temperature-dependence. In other words, the hot filament plays the role of a temperature sensor. By adding a known resistor in series, we are able to record voltages across both the device and the resistor (which gives the intensity) and determine the resistance of the sample as a function of time. The temperature coefficient of the MWs electrical resistance has been calibrated. The device was inserted in a cryostat ensuring isothermal experimental conditions. The electrical resistance was measured as the cryostat chamber temperature was slowly heating up to room temperature. Assuming temperature-dependence of the resistance to be $R(T) = R(T_0)[1 + \alpha(T - T_0)]$, the calibration leads to $\alpha = 1.98 \cdot 10^{-3} \Omega/K$ for $T_0 = 295 \text{ K}$.”

and in the main body as well with:

“Fig. 2a shows the emitted power detected by a fast MWIR detector (MCT, Kolmar) and the simulated signal using the measured temperature of the metallic wires (see Methods). [...] The observed delay in the rising phase, also observed in the temperature measurement, is attributed to thermal inertia of the SiN_x layer lying under the MWs.”

Besides, we have added its discussion in terms of maximum temperature and maximum power contrast in the legend of Fig. 2a:

“a, Emission pulse when applying a gaussian voltage. The FWHM of the voltage (blue) and emitted (orange) pulses are respectively 10 μs and 6.7 μs . The voltage is normalized by its maximum value 9.7 V, the measured optical signal is normalized by its maximum value (corresponding to 4.4 μW total emitted power). The optical signal was simulated (black dots) by computing $I_{BB}[\lambda_0, T_0 + \Delta T(t)] - I_{BB}[\lambda_0, T_0]$ where $\lambda_0 = 5.1 \mu\text{m}$, $T_0 = 295 \text{ K}$ is the ambient temperature and $\Delta T(t)$ is the measured temperature increase of the metallic wires (not shown, see Methods for calibration details). The maximum temperature $T_0 + \Delta T(t)$ used for normalization of the simulated optical signal is 570 K, for which a maximum power contrast of 4.4 μW is achieved.”

Reviewer #2 (Remarks to the Author):

The authors have answered all my questions. I have only one final suggestion. In the answer regarding the $1/\nu(\omega_{th})$ behavior, the author points out that this is valid for $D/\nu_{th} \sim \Lambda^{<2/sup>$, which in this case is beyond the detector cut-off. Still, this is an important condition (useful for device design) and should be added to the method section (line ~360).

We thank the referee for this suggestion. We have added the following in the Methods section :

“Note that the $1/\sqrt{\omega_{th}}$ drop in the frequency response is predicted by this model provided that the periodic array of MWs can be modelled as a uniform plane. This assumption is valid up to a modulation frequency $\nu_{th} \sim D/\Lambda^2 \sim 10^7$ Hz where Λ is the period of the structure and D the thermal diffusivity of SiN_x which is beyond our detector’s cutoff.”

Reviewer #3 (Remarks to the Author):

I thank the authors for comprehensively addressing the points raised by all the referees. My recommendation is that the paper is now published in Nature Communications.

We thank the referee for her/his positive review.